# Bridging Oceans and Atmosphere: A More Comprehensive Weather Model

## Abstract

The Earth system synthesizes multivariate interactions across atmospheric processes, oceanic circulations, cryospheric dynamics, and radiative forcing. While machine learning has transformed weather prediction within individual domains, current models inadequately capture cross-component couplings critical for holistic Earth system modeling. Fundamental challenges emerge from the disparities in spatial and temporal scales and prohibitive computational costs of training integrated models ab initio. To overcome these limitations, we present the Coupled Ocean-Atmosphere Framework (COAF), a deep learning architecture that dynamically couples pre-trained atmospheric and oceanic models through continuous-time modeling and a one-dimensional flux generation module to resolve scale mismatches. COAF introduces two pivotal innovations: An effective structure for cross-domain information exchange, analogous to energy-momentum transfer in physical models, which avoids structural overhauls of existing models, and an online replay buffer mechanism that drastically reduces memory consumption for long-term training. Experimental results demonstrate COAF's effectiveness in medium-range weather forecasting, achieving a 10% reduction in latitude-weighted RMSE for key prognostic variables (Z500, T2m) beyond 10-day forecast horizons. These advancements establish a new paradigm for coupled Earth system modeling.

## 1 Introduction

Recently, data-driven methods for weather forecasting, such as deep learning and machine learning (Kurth et al., 2023; Bi et al., 2023; Lam et al., 2022; Chen et al., 2023c; Kochkov et al., 2023; Nguyen et al., 2023), have emerged as mainstream tools in meteorology (Ben-Bouallegue et al., 2023; Ling et al., 2024). These methods have significantly advanced weather prediction across various domains, including atmospheric (Fig. 1(**Left**)) and oceanic systems (Fig. 1(**Middle**)). However, the coupling effects between these systems remain an open challenge. For instance, atmospheric processes involve the exchange of energy, water, and other substances with underlying surfaces (Oort & Rasmusson, 1971). Among these, the ocean, as the largest interface with the atmosphere, has a profound influence on atmospheric dynamics (Neukermans et al., 2018).

Despite its importance, effectively incorporating ocean data into atmospheric models remains challenging (Irrgang et al., 2021). Ocean datasets are comparable in size to atmospheric datasets, but integrating all relevant variables into a unified model is computationally prohibitive. Furthermore, a fundamental challenge arises from the temporal inconsistency between the two systems. Ocean data, typically available at daily or monthly intervals (Jean-Michel et al., 2021), exhibits slower variability compared to atmospheric data, which is often collected at hourly or six-hourly resolutions. This mismatch in temporal scales complicates the synchronization of datasets and hinders the development of coupled models.

Traditional dynamical models (Ly, 1995) address this issue by characterizing ocean-atmosphere interactions through fluxes, such as heat flux, radiation flux, and moisture flux. These models leverage partial differential equations and flux information to integrate future atmospheric states over time, effectively mitigating the impact of temporal inconsistencies. However, current AI-based approaches struggle to replicate this synergy. Most existing methods either: (1) oversimplify oceanic dynamics by focusing solely on surface-level variables such as sea surface temperature (SST), neglecting

critical subsurface processes (Zhong et al., 2024); or (2) downsample atmospheric data to match the temporal resolution of oceanic data, as seen in models like OLA (Wang et al., 2024), which discard essential high-frequency signals. Furthermore, the rapid development of AI solutions (Bodnar et al., 2024; Nguyen et al., 2024; Xu et al., 2024; Couairon et al., 2024; Esteves et al., 2023; Kochkov et al., 2024; Price et al., 2023; Han et al., 2024; Chen et al., 2025) for mid-range weather forecasting is noteworthy. Implementing a unified coupling framework could facilitate the evolution of existing AI models into comprehensive Earth system models.

To this end, we propose a Coupled Ocean-Atmosphere Framework (COAF), a novel approach designed to address the temporal inconsistency between oceanic and atmospheric processes while minimizing modifications to existing atmospheric models (Fig. 1(**Right**)). Our method leverages downsampled ERA5 reanalysis data (Hersbach et al., 2020) at six-hourly temporal resolution, combined with daily ocean reanalysis data (GLORYS12V1, (Jean-Michel et al., 2021)). COAF bridges the timescale discrepancy between rapid atmospheric dynamics and slower oceanic processes through three key strategies: (I) An autoregressive iterative method models the rapidly changing atmosphere, while a continuity-based approach simulates the gradual ocean, ensuring temporal coherence. (II) A flux generation module is designed that maps data into one-dimensional spatial features, along with a condi-

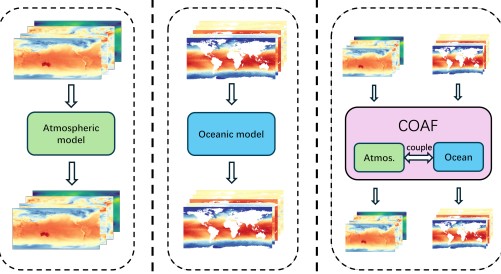

Figure 1: Schematic comparison of modeling paradigms. (**Left**) Previous atmospheric prediction models. (**Middle**) Previous oceanic prediction models, focusing on ocean dynamics. (**Right**) Our proposed coupled ocean-atmosphere model, integrates atmospheric and oceanic processes to address temporal inconsistencies and enhance forecasting accuracy.

tional control mechanism to facilitate interaction between the models. (III) Finally, a memory-efficient online replay buffer strategy enhances long-term forecasting performance while minimizing computational costs. Experimental results demonstrate that COAF reduces the latitude-weighted root-mean-square error (WRMSE) by approximately **10%** for key atmospheric variables, such as Z500 and T2m, when the forecast lead time exceeds 10 days.

Key contributions of this study include:

- **A Novel Plug-and-Play Coupling Framework:** We propose COAF, a generalizable framework that efficiently upgrades existing pre-trained atmospheric models into coupled ocean-atmosphere systems. It resolves the key challenge of temporal scale mismatch through a hybrid auto-regressive and continuous-time modeling approach.
- **An Efficient Conditional Interaction Mechanism:** We introduce a learned flux-like vector to enable cross-system interaction, which avoids costly architectural modifications to the pre-trained models and eliminates the need for training models from scratch.
- **A Memory-Efficient Training Strategy:** We propose an online replay buffer that reduces the memory for long-term fine-tuning by over four orders of magnitude (from 4.6 GB to 0.08 MB), enabling the training of large-scale, high-resolution coupled models.

## 2 RELATED WORK

**Deep Learning-based Weather Forecasting.** Traditional numerical weather prediction (NWP) has long relied on physical simulations of atmospheric dynamics. The paradigm began shifting with the introduction of machine learning approaches, catalyzed by the WeatherBench benchmark (Rasp et al., 2020). Some methods (Weyn et al., 2020; Hu et al., 2023; Nguyen et al., 2023; Esteves et al., 2023) employed deep neural networks such as ResNet (He et al., 2016), U-net (Ronneberger et al., 2015), Swin Transformer (Liu et al., 2021) and so on to enhance model performance on this dataset, but the performance of these data-driven models still has a big gap to The IFS-HRES.

In response, Chen et al propose SwinRDM (Chen et al., 2023b) by integrating an improved Swin-VRNN (Hu et al., 2023), which achieves higher performance than IFS at lead times of up to 5 days at 1.4° in some variables. Subsequently, several researchers (Bi et al., 2023; Lam et al., 2022; Chen et al., 2023a;c; Price et al., 2023; Couairon et al., 2024) developed advanced machine learning models that significantly enhance weather forecasting, surpassing the traditional IFS-HRES method in

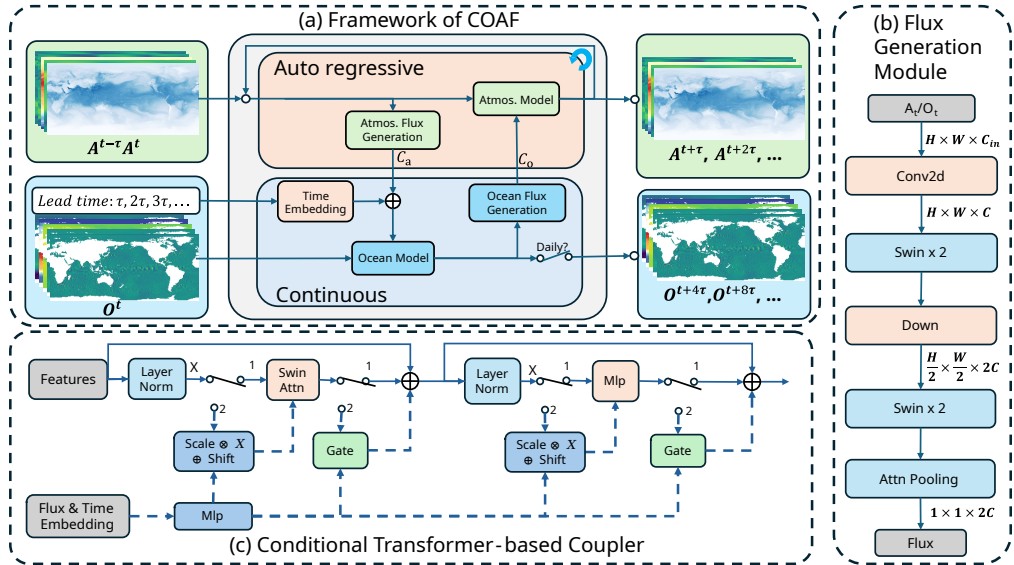

Figure 2: **Overview of the Coupled Ocean-Atmosphere Framework (COAF).** (a) A dual-pathway framework coupling autoregressive atmospheric and continuous-time oceanic models via flux vectors ($C_a$, $C_o$). (b) The module that encodes spatial states into 1D flux vectors. (c) The coupler that uses external signals for conditional feature modulation.

predictive performance at higher resolution ($0.25°$). However, these models are limited to training with only atmospheric variables, ignoring interactions across layers like ocean-atmospheric coupling. In this paper, we propose COAF, which couples oceanic data into global weather forecasting and achieves further performance enhancements.

**Ocean-Atmosphere Couple Dynamics.** Traditional numerical weather prediction models have already taken into account the complex interplay between different Earth systems (Srinivas et al., 2016; Rainaud et al., 2017). These models commonly employ a coupler approach (Liu et al., 2018) to integrate atmospheric and other models. In this process, the coupler acts as an intermediary, facilitating the exchange of data and information between models. This bidirectional interaction enhances the accuracy of medium- to long-term weather forecasts by incorporating dynamic feedback from both systems (Vellinga et al., 2020).

Currently, Zhong et al. propose FuXi-ENS (Zhong et al., 2024), which includes one ocean variable (sea surface temperature, SST) and employs an ensemble strategy to attempt probabilistic weather forecasting. Although they consider the influence of the ocean, SST is not enough for a complete ocean-atmosphere forecasting system. Signals from deeper ocean, like multi-layer ocean currents, are also necessary (Vellinga et al., 2020; Rainaud et al., 2017). In contrast, more recently models like OLA (Wang et al., 2024) and Dlesym (Cresswell-Clay et al., 2025) employ complete a ocean model to simulate the whole atmosphere and ocean system, but focus on the seasonal climate prediction.

Despite the utilization of both oceanic and atmospheric variables for seasonal climate prediction, the impact of the ocean on global weather forecasts at the medium-term scale remains unexplored. This paper proposes COAF, modeling the ocean and atmospheric modules separately and achieving coupling for medium-term forecasting across different time scales through conditional control.

## 3 METHODS

### 3.1 PROBLEM FORMULATION

Traditional dynamical models (Ly, 1995) implement ocean-atmosphere coupling through flux-based differential equations:

$$\mathcal{F}_t^{A2O}, \mathcal{F}_t^{O2A} = F_{\text{couple}}(\mathbf{A}_t, \mathbf{O}_t), \tag{1}$$

$$\mathbf{A}_{t+\tau} = F_{\text{atm}}(\mathbf{A}_t, \mathcal{F}_t^{O2A}, \tau), \tag{2}$$

$$\mathbf{O}_{t+\tau} = F_{\text{ocn}}(\mathbf{O}_t, \mathcal{F}_t^{A2O}, \tau). \tag{3}$$

Here, $\text{F}_{\text{couple}}$ denotes the flux parameterization operator calculating. The state transition operators $\text{F}_{\text{atm}}$ and $\text{F}_{\text{ocn}}$ govern atmospheric and oceanic evolution, respectively, with $\tau$ representing the discrete integration timestep. $\mathcal{F}_t^{A2O}$ denotes atmosphere-to-ocean fluxes (eg. momentum, heat). $\mathcal{F}_t^{O2A}$ represents ocean-to-atmosphere fluxes (eg. moisture, radiation). Due to the inherent coupling between these systems, the intermediate variables generated through their interaction are often not directly observable, which consequently leads to the incorporation of numerous empirical parameters in the parameterization formulas.

To overcome the empirical limitations of physical parameterization while retaining first-principles coupled logic, we believe that COAF should have the following key components:

- **Flux Generation**: The module employs domain-specific feature extractors to directly derive flux representations from oceanic and atmospheric states, respectively, through separate deep learning encoders. These data-driven flux embeddings serve as controlling signals for cross-system state transitions, preserving dynamical coupling mechanisms akin to traditional flux-exchange formulations while eliminating explicit physical parameterization.
- **Atmosphere Evolution**: The atmospheric predictor module generates subsequent states $\mathbf{A}_{t+\tau}$ through autoregressive modeling of current atmospheric conditions $\mathbf{A}_{t-\tau}, \mathbf{A}_t$ and cross-system coupling features $\mathbf{C}_{o,t+\tau}$ from the Flux Generation module, ensuring accurate atmospheric forecasts.
- **Ocean Evolution**: The oceanic predictor employs continuous-time neural operators to model state evolution $\hat{\mathbf{O}}_{t+\tau}$ over arbitrary intervals $\tau \in \mathbb{R}^+$, conditioned on dynamically interpolated atmospheric coupling signals $\mathbf{C}_{a,t}$.

To meet these requirements, COAF defines four key functions. These functions are computed sequentially to enable comprehensive ocean-controlled weather forecasting:

$$\mathbf{C}_{a,t} = \text{F}_e^a \left( \mathbf{A}_{t-\tau}, \mathbf{A}_t, \theta_e^a \right), \tag{4}$$

$$\hat{\mathbf{O}}_{t+\tau} = \text{F}_o \left( \mathbf{O}_t, \tau, \mathbf{C}_{a,t}, \theta_o \right), \tag{5}$$

$$\mathbf{C}_{o,t+\tau} = \text{F}_e^o \left( \hat{\mathbf{O}}_{t+\tau}, \theta_e^o \right), \tag{6}$$

$$\hat{\mathbf{A}}_{t+\tau} = \text{F}_a \left( \mathbf{A}_{t-\tau}, \mathbf{A}_t, \mathbf{C}_{o,t+\tau}, \theta_a \right). \tag{7}$$

**Eq. 4** ($\text{F}_e^a$): Extracts atmospheric interaction features $C_{a,t}$ from the current and previous atmospheric states $(A_t, A_{t-\tau})$, using learned parameters $\theta_e^a$. **Eq. 5** ($\text{F}_o$): Predicts oceanic state $\hat{O}_{t+\tau}$ by integrating initial ocean state $O_t$, timestep $\tau$, and the atmospheric features $C_{a,t}$ with parameters $\theta_o$. **Eq. 6** ($\text{F}_e^o$): Derives oceanic coupling features $C_{o,t+\tau}$ from predicted state $\hat{O}_{t+\tau}$ using parameters $\theta_e^o$. **Eq. 7** ($\text{F}_a$): Generates atmospheric state $\hat{A}_{t+\tau}$ by fusing historical states $(A_t, A_{t-\tau})$ with oceanic features $C_{o,t+\tau}$ via parameters $\theta_a$.

## 3.2 The Flowchart of COAF

The flowchart of COAF is illustrated in Figure 2 The training process facilitates a bidirectional information exchange between the ocean and atmosphere modules. The sequence begins with the Atmospheric Flux Generation Module, which takes the current and previous atmospheric states as input to produce a compact feature representation—the atmospheric flux vector. This vector is then fused with the target time embedding to serve as a conditional signal for the Ocean Model, steering its prediction of the future oceanic state. Subsequently, the process reverses. The predicted oceanic state is fed into the Oceanic Flux Generation Module to derive an analogous oceanic flux vector, which then acts as the conditional input for the Atmosphere Model, guiding its subsequent forecast.

## 3.3 Flux Generation Module

The Flux Generation module is designed to encode the high-dimensional state of one Earth system (e.g., the ocean) into a compact, one-dimensional vector. This vector serves as a learned latent representation, analogous to physical fluxes in traditional dynamical systems, that provides a conditional signal to control the evolution of the other system (e.g., the atmosphere). As depicted in Figure 2(b), the process begins with a convolutional network using a $1 \times 1 \times C$ kernel to extract features into a size of $H \times W \times C$. This is followed by a Swin Transformer and downsampling module for spatial feature fusion, reducing the dimensions to $H/2 \times W/2 \times 2C$. Attention pooling is then applied to

condense these features to a one-dimensional scale, specifically $1 \times 1 \times 2C$. This output serves as a conditioning input for the ocean prediction model, allowing the next ocean prediction to incorporate the current atmospheric state. The predicted ocean data is then reprocessed through the same module to extract comprehensive information, further refining the atmospheric autoregressive iterative predictions.

### 3.4 THE "ENCODE-FUSE-DECODE" STRUCTURE.

The atmosphere and ocean models in our framework both follow the "encode-fuse-decode" structure (Chen et al., 2023a; Bodnar et al., 2024) (See Supplementary A.2). Here, we made two modifications to adapt this structure to ocean-controlled atmospheric modeling.

**Classification of Model Inputs.** Atmospheric variables are previously considered as different classes (Chen et al., 2023a; Bodnar et al., 2024) and assigned to different expert models, i.e., encoder-decoder pairs. Following prior conventions, we adopt six atmospheric classes: geopotential, temperature, humidity, zonal/meridional winds, and surface pressure.

For oceanic variables, however, we adopt a depth-based classification approach, assigning different depth levels to distinct expert models. This shift from variable-based to depth-based classification is due to the variation in modeling errors across ocean layers (Gouretski, 2018) as well as the increasing sparsity of data in deep ocean. Based on the stratification of ocean layers (Li et al., 2020), we roughly divide ocean depths as follows: (0-5m, 5-16m, 16-56m, 56-187m, 187-1063m), resulting in five modalities, which contain 5, 8, 12, 12, and 16 variables, respectively.

**Time Encoding**. Due to the ocean's high density and heat capacity, changes also occur slowly (Hoegh-Guldberg et al., 2014), so ocean data are collected every day rather than every 6 hours like atmospheric data. To facilitate the coupling of temporal and atmospheric models, we have formulated ocean prediction as a continuous forecasting model. As shown in the figure 2(c), we adopt the time encoding method used in DiT (Peebles & Xie, 2023), applying sine and cosine encoding to time $t$ as a condition to control the output of each transformer block in the ocean prediction model. Specifically, leveraging the current oceanic state and a forecast time interval $t$, we generate the oceanic state at time $t$ as per the formula provided. An essential advantage of this approach lies in training with daily time intervals, enabling the generation of predictions at any time interval $t$ during inference. This feature streamlines the alignment of time scales in ocean-atmosphere coupled model training and inference with atmospheric prediction models.

### 3.5 CONDITIONAL TRANSFORMER-BASED COUPLER

To enable the interaction between the ocean and atmosphere models, we modulate the features of the atmospheric model using the learned flux vector from the ocean model. This is achieved through an adaptive feature modulation mechanism, a technique proven effective in other domains for injecting conditional information, such as in style transfer (AdaIN) (Huang & Belongie, 2017) and conditional image synthesis(Zhan et al., 2024). Specifically, as shown in Figure 2(c), each block in our atmospheric model's transformer can operate in two modes, controlled by a conceptual switch. When the switch is set to **"1"**, the transformer is degraded to the traditional Swin-Transformer (Liu et al., 2021). Conversely, when the switch is set to **"2"**, conditional inputs will be considered and interactions between ocean and atmosphere are involved. To achieve such comprehensive ocean-atmospheric coupling, conditional Swin-Transformer is trained in three stages.

**Stage 1: atmosphere module training.** As shown in Figure 2(c), the atmosphere model is trained with the switch set to **"1"**. At this stage, only atmospheric variables are considered, and the transformer is reduced to a traditional Swin-Transformer, as used in previous weather forecasting models (Bi et al., 2023; Chen et al., 2023a).

**Stage 2: ocean module training.** Unlike the atmosphere module, the ocean module is trained with the switch set to **"2"**. Conditional inputs include time embedding features, as the ocean is formulated as a continuous time forecasting model due to its slower pace of change compared to the atmosphere, as we have discussed.

**Stage 3: ocean-atmosphere coupling.** After independently training the atmosphere and ocean modules, an additional training phase is required to couple them, enabling ocean features to condition

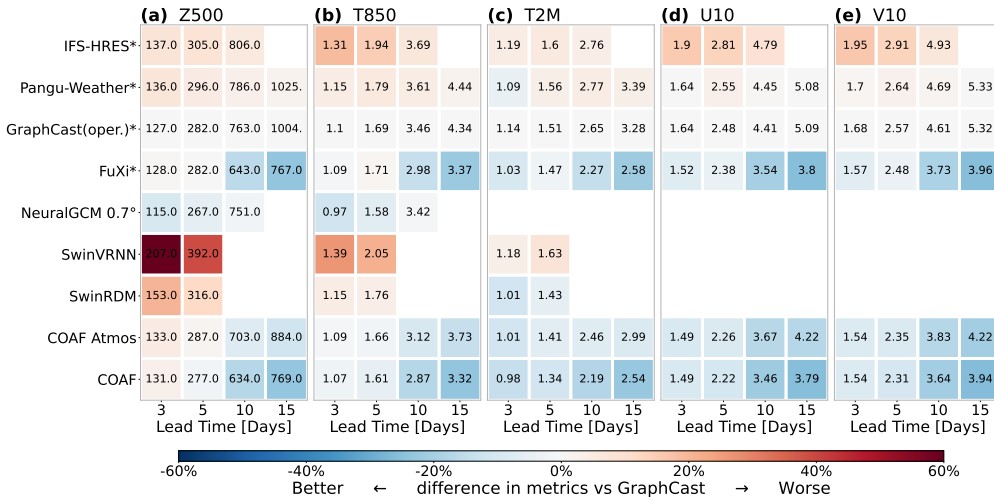

Figure 3: Comparison of different models on WRMSE (lower is better) at lead times of 3, 5, 10, and 15 days. The models marked with * were trained and tested using data with a resolution of 0.25°. NeuralGCM 0.7° utilized data at 0.7° resolution, and the remaining models used 1.4° resolution.

the atmospheric evolution and establish ocean-atmosphere interactions. In this phase, the switch in the atmosphere model is set to **"2"**. The parameters of the atmospheric Flux Generation Module are initialized to ensure that the initial atmospheric feature output is zero.

Additionally, for a stable start to coupled fine-tuning, the parameters of the conditional control module (Figure 2(c)) are specifically initialized: the modulating scale, shift, and gate vectors are set to 1, 0, and 1, respectively. This initialization ensures that, at the beginning of training, the atmospheric and oceanic models function independently. As training progresses, the parameters adjust, gradually incorporating ocean-atmosphere interactions.

### 3.6 ONLINE REPLAY BUFFER

The replay buffer training strategy (Details in Supplementary A.4) has achieved significant success in enhancing the long-term performance of weather forecasting (Bodnar et al., 2024), allowing for gradually increasing the training probability for longer forecast steps as training progresses. At the same time, it ensures that data with shorter forecast steps has a higher training probability to maintain short-term performance while enhancing long-term prediction accuracy. In this study, we adopt a probability simulation method to model the autoregressive training step probabilities.

As shown in Supplementary Algorithm 3. The autoregressive iteration steps $n$ during the training period are obtained by the online replay buffer algorithm (Supplementary Algorithm 2). Specifically, when the algorithm returns a training step $n$, it indicates that we need to train the data sequentially from the first step to the nth step. Compared to the replay buffer algorithm in Previous Work (Chen et al., 2023a; Bodnar et al., 2024), this method effectively saves memory.

To achieve this, we set up a list, $C_{train}$, to record how many times each training step has been trained. When sampling a training step, we first check if the sampled value is greater than or equal to the training count. If it is less, it means we have already trained the data for that step in previous samples. Otherwise, we proceed to train the data for that step. By using this method, we achieve sampling probabilities similar to the replay buffer strategy while significantly reducing memory usage by avoiding the need to store intermediate variables.

## 4 EXPERIMENT

### 4.1 EXPERIMENTAL SETUP

**Tasks.** We evaluate COAF's performance on a 1.4° global, multi-level, medium-range weather forecast through a deterministic model to predict the next-step atmospheric ($A_{t+\Delta t}$) and oceanic variables ($O_{t+\Delta t}$). These predictions are conditioned on the current and previous atmospheric states ($A_t, A_{t-1}$) as well as the initial oceanic state ($O_0$). The time step $\Delta t$ is set to 6 hours for atmospheric

variables. Given the differing update frequencies of the oceanic data (once per day) compared to the atmospheric data (every 6 hours), oceanic predictions are updated every four atmospheric prediction time steps. (Details, please refer to Supplementary A.5).

**Datasets.** In this study, atmospheric data was sourced from ERA5 (Hersbach et al., 2020). And we selected 4 surface variables and 5 multi-level variables, along with 13 pressure levels. This brings the total to 69 variables. For the ocean data, we utilized the GLORYS12V1 (Jean-Michel et al., 2021), developed within the framework of the Copernicus Marine Environment Monitoring Service (CMEMS). This product provides daily mean states of multi-layer ocean data. To align with the atmospheric data, we selected temperature, salinity, the U and V component of currents, and sea surface height, with data taken from 13 vertical levels. Details in Supplementary Table 4.

To manage computational requirements, both datasets were downsampled to a 1.4° resolution using bilinear interpolation. While the state-of-the-art in data-driven forecasting has converged on a higher 0.25° resolution, the primary contribution of this work is a novel coupling framework. We therefore adopt the 1.4° resolution as a standard benchmark to rigorously isolate and demonstrate the significant performance gains attributable to ocean-atmosphere coupling. For the atmospheric model's pre-training (Stage 1), we selected the period from 1979 to 2015. The subsequent training phases (Stages 2 and 3) were aligned to use the same data periods, with the years 1993 to 2021 serving as the training set and 2022 designated as the test set.

**Metrics.** Consistent with previous works (Bi et al., 2023; Lam et al., 2022; Hu et al., 2023; Chen et al., 2023b), we used latitude-weighted root-mean-square error (WRMSE) and Anomaly Correlation Coefficient (WACC) to evaluate forecast performance.

## 4.2 TRAINING HYPERPARAMETER

**Stage 1: atmosphere model training.** The atmospheric prediction model uses two time steps of atmospheric states as input to forecast the next time step, which utilizes the AdamW optimizer with a batch size of 32 and is trained for 50 epochs. Additionally, a cosine annealing schedule is implemented, with an initial learning rate of 5e-4, and gradually decreasing it to zero.

**Stage 2: ocean model training.** In contrast to atmospheric model training, the ocean prediction model uses one step of ocean state and target time as the input conditions during the training period. The other training hyperparameters are set the same as those in atmospheric model training.

**Stage 3: ocean-atmosphere coupling.** Finally, we integrate the pre-trained models into COAF, then use the online replay buffer strategy to train the overall model with a batch size of 8 for 5 epochs, each containing 10,000 steps. The training process utilizes the AdamW optimizer with a cosine annealing schedule, starting with an initial learning rate of 5e-6.

## 4.3 EXPERIMENT RESULTS

**Compared Methods.** To evaluate the forecast skills of COAF, we conducted a comparative analysis, comparing its performance with the state-of-the-art dynamical model (IFS-HRES) and leading AI methods in medium-range weather forecasting (Figure 3). The forecast skill scores for IFS-HRES were sourced from its official 2022 dataset repository. For the SwinVRNN, SwinRDM, and Neural-GCM 0.7° models, we refer to performance metrics reported in their original research papers. Unfortunately, neither SwinVRNN nor SwinRDM considers the deterministic weather forecast that exceeds 5 days. To further validate our model's effectiveness at longer lead times, we compared COAF with high-impact AI weather forecasting models, including Pangu-Weather, GraphCast(oper.)[1], and FuXi, all of which operate at a 0.25° spatial resolution. The performance for these models was obtained using their open-source frameworks on the 2022 dataset.

**Performance of COAF.** Figure 3 and Supplementary Table 8 indicate that our model performs well in terms of WRMSE and WACC, particularly for surface variables, where it consistently outperforms previous models. It has reduced the WRMSE by 5-12% compared with SwinRDM at a 5-day lead time. Moreover, our model remains competitive over the 5-day lead time, outperforming current state-of-the-art high-resolution models.

---

[1]GraphCast(oper.) is the official model fine-tuned on HRES data from 2016-2021, its inference results shown here are based on the operational analysis dataset.

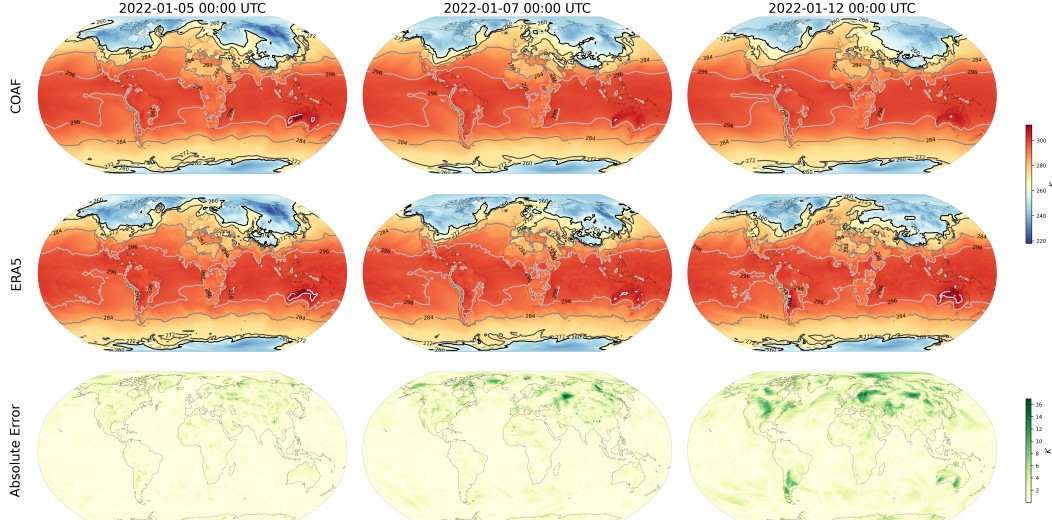

Figure 4: Visualization of t2m prediction results for 3-Day, 5-Day, and 10-Day forecasts at the initialization time of January 2, 2022, 00:00 UTC.

To qualitatively evaluate the forecast quality, Figure 4 and Supplementary Figure 6 visualize the prediction results for T2m and Z500 at 3, 5, and 10-day lead times. The forecasts (top row) maintain high fidelity to the ERA5 ground truth (middle row), accurately capturing the large-scale patterns and evolution of weather systems. The error analysis (bottom row) reveals that forecast errors occur in the mid-to-high latitudes, regions known for complex and frequent interactions between cold and hot air masses.

Table 1: Training efficiency comparison of different weather models. The table shows the hardware platform, number of devices, and training time for COAF, Pangu, GraphCast, and Stormer. The training cost for COAF is broken down into its three main stages.

| Method | Hardware | Number | Time |
| --- | --- | --- | --- |
| Pangu | Nvidia V100 | 192 | 16d |
| GraphCast | TPU v4 | 32 | 28d |
| Stormer | Nvidia A100 | 128 | 24h |
| COAF Stage 1 | Nvidia A100 | 8 | 2.5d |
| COAF Stage 2 | Nvidia A100 | 8 | 14h |
| COAF Stage 3 | Nvidia A100 | 8 | 20h |

**Ablation Study on Coupling Ocean-Atmosphere.** To demonstrate the necessity of ocean-atmosphere coupling, we performed a comprehensive ablation study by comparing our full COAF model against a variant trained without the ocean module (COAF Atmos). As shown in Figure 3, the incorporation of oceanic data yields a consistent performance improvement across all variables and forecast horizons. The benefits become particularly pronounced at longer lead times; after 10 days, the coupled model achieves a ~10% reduction in WRMSE for Z500, T850, and T2M, and a ~5% reduction for U10 and V10.

Furthermore, to validate the versatility of our framework, we replaced our atmospheric backbone with the Pangu-Weather model and retrained the coupling modules. The resulting coupled model again showed significant performance gains (see Supplementary B.4), proving that COAF is a generalizable framework that can enhance other state-of-the-art atmospheric models.

**Computational Efficiency and Framework Flexibility.** A key advantage of the COAF framework lies in its exceptional computational efficiency, particularly in its "plug-and-play" capability to enhance existing pre-trained models. As detailed in Table 1, while Stages 1 and 2 represent a one-time pre-training cost, the most important coupled fine-tuning (Stage 3) is remarkably rapid, requiring only 20 hours on 8 NVIDIA A100 GPUs. This efficiency highlights the framework's significant practical value and flexibility. It enables researchers to easily integrate a powerful pre-trained atmospheric model—which may have required thousands of GPU-days to train—into a more physically consistent and accurate coupled ocean-atmosphere system with minimal additional computational cost. This demonstrates that COAF is not merely a standalone model, but an efficient and versatile framework for facilitating ocean-atmosphere coupling.

**Ablation Study on Ocean Prediction Model.** Due to the slower dynamics of the ocean compared to the atmosphere, we employ a continuous-time modeling approach tailored for oceanic systems. Considering the varying physics at different depths, we adopt a multi-expert design that treats distinct

Table 2: Comparison of different ocean prediction models at lead times of 1, 5, 10, and 15 days for Surface Temperature, V component of currents and Ocean salinity in 2022.

| Method | thetao | vo | so |
|---|---|---|---|
| FourcastNet | **0.13**/2.95/4.06/4.97 | **0.037**/0.183/0.260/0.314 | **0.072**/1.047/1.450/1.634 |
| COAF Ocean iter | **0.13**/1.25/2.37/3.52 | 0.038/0.128/0.180/0.216 | 0.073/0.862/1.321/1.629 |
| COAF Ocean* | 0.14/0.36/0.49/0.53 | 0.042/0.089/0.105/0.114 | 0.077/0.176/0.217/0.238 |
| **COAF Ocean** | **0.13**/0.35/0.46/0.52 | **0.037**/**0.080**/**0.097**/**0.106** | 0.073/**0.171**/**0.211**/**0.233** |
| Climatology | 1.94/1.94/1.94/1.94 | 0.133/0.133/0.133/0.133 | 0.513/0.513/0.513/0.513 |

Table 3: Performance comparison of Replay Buffer and Online Replay Buffer. The latitude-weighted RMSEs are shown for key variables at lead times of 3, 5, and 10 days, alongside the required memory. Results are evaluated on 2022 ERA5 data.

| Method | Z500 | T850 | T2M | Memory |
|---|---|---|---|---|
| Replay Buffer | 132/281/679 | 1.08/1.63/3.04 | 0.99/1.37/2.33 | 4692.19M |
| Online Replay Buffer | **131/277/634** | **1.07/1.61/2.87** | **0.98/1.34/2.19** | **0.08M** |

ocean layers as separate modalities, detailed in Section 3.4. To validate our architectural designs, we conduct two ablation studies with the latitude-weighted RMSE presented in Table 2. First, we compare our continuous-time approach against standard autoregressive methods, including FourcastNet and the iterative version of our model (COAF Ocean iter). Second, we evaluate our depth-based expert strategy by comparing our final model, COAF Ocean, against a variant that partitions experts by variable instead of depth, denoted COAF Ocean*.

While autoregressive models (FourcastNet, COAF Ocean iter) perform competitively on the first day, they suffer from rapid error accumulation at longer lead times. In contrast, our continuous-time models maintain significantly lower error growth. Notably, the depth-based classification COAF Ocean consistently outperforms the variable-based classification COAF Ocean*, confirming the superiority of our proposed multi-expert, continuous-time architecture for stable and accurate long-range ocean forecasting.

**Effectiveness of Online Replay Buffer.** The Online Replay Buffer is designed to mitigate the prohibitive memory overhead of the standard replay buffer strategy by simulating its training effects without explicitly storing intermediate model outputs. The effectiveness of this approach is demonstrated in Table 3. Our online strategy not only achieves slightly better long-term forecasting accuracy across key variables (Z500, T850, and T2M) but also reduces memory consumption by over four orders of magnitude—from over 4.6 GB to a negligible 0.08 MB. These results were obtained using 69-channel data at 1.4° resolution with a buffer length of $200^2$. Most importantly, while the memory of a standard replay buffer scales linearly with buffer size and data resolution, our online strategy is unaffected by this limitation, making it a far more scalable solution for high-resolution models.

## 5 DISCUSSION AND FUTURE WORK

In this paper, we introduce the Coupled Ocean-Atmosphere Framework (COAF), an AI-driven approach designed to achieve temporal consistency between oceanic and atmospheric predictions. By employing a conditional control method for data integration and an online replay buffer for memory-efficient long-term training, our framework creates a more cohesive and robust forecasting system. Our results demonstrate that COAF establishes a new state-of-the-art for coupled ocean-atmosphere forecasting at the standard 1.4° resolution, providing a crucial proof-of-concept for our coupling methodology and training strategy. Looking ahead, the immediate priority is to scale the COAF framework to the operational 0.25° resolution. Furthermore, the flexible nature of our framework opens exciting avenues for incorporating other Earth system components, such as sea ice or land surface models, paving the way for more holistic AI-based Earth system simulations.

---

[2]A queue of length 200 stores atmospheric data for two time steps, oceanic data for one time step (in float32), and target indices. The required storage is approximately: $\frac{200 \times (2 \times 69 \times 128 \times 256 + 53 \times 120 \times 256) \times 4}{1024 \times 1024} \approx 4692$, MB

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

# A    METHOD

## A.1    THE "ENCODE-FUSE-DECODE" STRUCTURE.

The atmosphere and ocean models in our framework both follow the "encode-fuse-decode" structure in general (Figure 5). Initially, multi-dimensional data is classified and assigned to different expert models, i.e., encoder-decoder pairs. This classification process resembles a Mixture-of-Experts approach but is predetermined by criteria such as variable types or depth levels. The classified data is then embedded into a feature space using a convolutional network with a $1 \times 1 \times C$ kernel. Next, each data class is downsampled to $H/2 \times W/2 \times 2C$ using an encoder comprising two Swin Transformer blocks. The downsampled data is then concatenated in a feature fusion module, where attention mechanisms and MLPs integrate features across various positions. After fusion, the data is directed to the corresponding decoders, which upsample it to its original dimensions. To preserve high-resolution details, a residual connection is applied to each encoder-decoder pair, ensuring accurate predictions across diverse multi-dimensional variables.

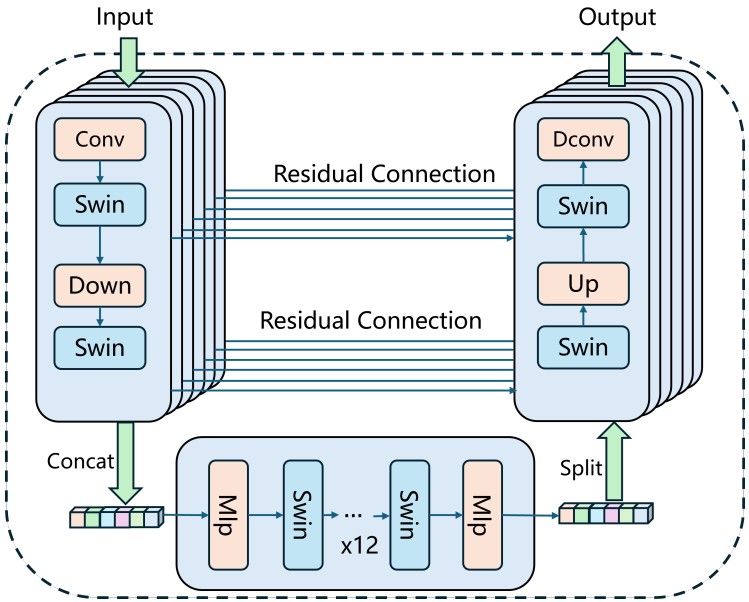

Figure 5: Overview of "encode-fuse-decode" structure

## A.2    ATMOSPHERE PREDICT MODEL

We employ an iterative approach to pre-train the atmospheric prediction model. Specifically, we use a six-hour interval method to predict the atmospheric state six hours ahead based on the current state and the state six hours earlier, shown in the following formula:

$$\hat{X}^{i+1} = \text{AtmosphereModel}\left(X^i, X^{i-1}\right), \tag{8}$$

$x^t$ is the current atmospheric state, $x^{t-1}$ denotes the state six hours ago, and $x^{t+1}$ represents the state six hours in the future. Considering the varying speed of changes in the numerical ranges of different variables, we utilize a probability loss, illustrated as:

$$\hat{\mu}^{i+1}, \hat{\sigma}^{i+1} = \text{AtmosphereModel}\left(X^i, X^{i-1}\right), \tag{9}$$

where $\hat{\mu}^{i+1}$ and $\hat{\sigma}^{i+1}$ are predicted mean and variance of prediction $X^{i+1}$. And the probability of atmosphere variables can be calculated according to $\hat{\mu}^{i+1}$ and $\hat{\sigma}^{i+1}$ of Eq. 9.

The model generates the mean and standard deviation for each variable at each position for the next atmospheric state.

$$p\left(x^{i+1}_{c,w,h} \mid \hat{\mu}^{i+1}_{c,w,h}, \hat{\sigma}^{i+1}_{c,w,h}\right) = \mathcal{N}\left(\hat{\mu}^{i+1}_{c,w,h}, \hat{\sigma}^{i+1}_{c,w,h}\right), \tag{10}$$

Table 4: Atmospheric and oceanic variables utilized in our datasets. The atmospheric multi-level variables are measured at 13 pressure levels: 50 hPa, 100 hPa, 150 hPa, 200 hPa, 250 hPa, 300 hPa, 400 hPa, 500 hPa, 600 hPa, 700 hPa, 850 hPa, 925 hPa, and 1000 hPa. The oceanic multi-depth variables correspond to depth levels: 0.49 m, 5.07 m, 15.81 m, 25.21 m, 40.34 m, 55.76 m, 109.73 m, 155.85 m, 186.13 m, 318.13 m, 453.94 m, 763.33 m, and 1062.44 m.

| Type | Variable name | Short name | level |
|---|---|---|---|
| | Geopotential | z | 13 |
| | Specific humidity | q | 13 |
| | U component of wind | u | 13 |
| Atmos. | V component of wind | v | 13 |
| | Temperature | t | 13 |
| | 10 metre u wind component | u10 | 1 |
| | 10 metre v wind component | v10 | 1 |
| | 2 metre temperature | t2m | 1 |
| | Mean sea level pressure | msl | 1 |
| | Ocean temperature | thetao | 13 |
| | Ocean salinity | so | 13 |
| Ocean. | U component of currents | uo | 13 |
| | V component of currents | vo | 13 |
| | Sea surface height | ssh | 1 |

where each element $x_{c,w,h}^{i+1}$ in $X^{i+1}$ with subscript $(c, w, h)$ follows an independent univariate Gaussian distribution $\mathcal{N}\left(\hat{\mu}_{c,w,h}^{i+1}, \hat{\sigma}_{c,w,h}^{i+1}\right)$, where $c = 1, \ldots, 69$ denotes the index for the channel, i.e. different pressure levels and weather variables. $w$ and $h$ respectively denote the latitude grid and longitude grid.

Optimizing the model by maximizing the probability of the actual state at the next moment falling within the Gaussian distribution composed of the current mean and standard deviation, as illustrated in Eq. 13. This method allows for the learning of weights through the standard deviation for different atmospheric variables at different positions.

$$\text{loss} = -\log E\left(p\left(x_{c,w,h}^{i+1} \mid \hat{\mu}_{c,w,h}^{i+1}, \hat{\sigma}_{c,w,h}^{i+1}\right)\right) \tag{11}$$

$$= -\log E\left(\frac{1}{\sqrt{2\pi(\hat{\sigma}_{c,w,h}^{i+1})^2}}\exp(-\frac{(x_{c,w,h}^{i+1} - \hat{\mu}_{c,w,h}^{i+1})^2}{2(\hat{\sigma}_{c,w,h}^{i+1})^2})\right) \tag{12}$$

$$= E(\frac{(x_{c,w,h}^{i+1} - \hat{\mu}_{c,w,h}^{i+1})^2}{2(\hat{\sigma}_{c,w,h}^{i+1})^2} + \log(2(\hat{\sigma}_{c,w,h}^{i+1})^2)) + C. \tag{13}$$

### A.3 OCEAN PREDICT MODEL

Ocean prediction is modeled as a continuous-time prediction process. Given the current oceanic state and a prediction time interval $t$, the model directly generates the predicted ocean state for day t, as shown in:

$$\hat{X}^{i+t} = \text{OceanModel}\left(X^i, t\right), \tag{14}$$

During pre-training, we optimize the model parameters by minimizing the mean absolute error (MAE) between the predicted results conditioned on any time interval $t$ and the target ocean state.

$$L(\theta) = E_{t \sim P(t), (X_0, X_t) \sim D}[\||f_\theta(X_0, t) - X_t\||_2], \tag{15}$$

where $P(t)$ represents the sampling probability for different time intervals. In this experiment, we utilize an inverse function to generate sampling probabilities for time intervals ranging from 1 to 15 days.

## A.4 REPLAY BUFFER

The replay buffer strategy has demonstrated effective performance in the long-term fine-tuning of models. As illustrated in Algorithm 1, during the fine-tuning phase, the model enhances its long-term performance by simultaneously training on both real data and model predictions in a proportional manner. However, this approach requires storing intermediate variables in memory, which leads to significant memory resource consumption.

---

**Algorithm 1** Standard Replay Buffer Training

---

**Require:** Dataset $\mathcal{D}$, model $M$, finetune rate $k$
1: Initialize Replay Buffer $\mathcal{B}$ with max size.
2: **procedure** TRAINANDUPDATE$(x, y)$
3:  $\hat{y} \leftarrow M(x)$                ▷ Forward pass
4:  $\mathcal{L} \leftarrow \text{Loss}(\hat{y}, y)$             ▷ Compute loss
5:  Update parameters of $M$ using $\nabla\mathcal{L}$    ▷ Backward pass & optimizer step
6:  **return** $\hat{y}$
7: **end procedure**

8: **for** each batch $(x_t, y_t)$ with target index $t$ from $\mathcal{D}$ **do**
9:  $\hat{y}_t \leftarrow$ TRAINANDUPDATE$(x_t, y_t)$      ▷ Perform standard training step
10:  **if** $t$ is not the final step in sequence **then**
11:   Add $(\hat{y}_t, t+1)$ to buffer $\mathcal{B}$
12:  **end if**
13:  **for** $i = 1$ to $k$ **do**           ▷ Perform $k$ replay/finetuning steps
14:   $(x_j, t_j) \leftarrow$ Randomly sample from $\mathcal{B}$
15:   $y_j \leftarrow$ Get target from $\mathcal{D}$ at index $t_j$
16:   $\hat{y}_j \leftarrow$ TRAINANDUPDATE$(x_j, y_j)$
17:   **if** $t_j$ is not the final step in sequence **then**
18:    Add $(\hat{y}_j, t_j + 1)$ to buffer $\mathcal{B}$
19:   **end if**
20:  **end for**
21: **end for**

---

**Algorithm 2** Online Replay Buffer Logic

---

**Require:** Sample rate $k$, Deque length 'deque_len', Max forecast steps $T_{max}$
1: Initialize sample counts $C_{sample}[t] \leftarrow 0$ for $t = 1, \ldots, T_{max}$
2: Initialize training counts $C_{train}[t] \leftarrow 0$ for $t = 1, \ldots, T_{max}$
3: Initialize a deque $\mathcal{Q}$ with maximum length 'deque_len'.
4: **while** training **do**
5:  Add step '1' to $\mathcal{Q}$
6:  **if** $C_{sample}[1] \geq C_{train}[1]$ **then**
7:   $C_{train}[1] \leftarrow C_{train}[1] + 1$
8:   **yield** 1              ▷ Yield step 1 to be trained
9:  **end if**
10:  **for** $i = 1$ to $k$ **do**
11:   $t_s \leftarrow$ Randomly sample a step from $\mathcal{Q}$
12:   Add step $t_s + 1$ to $\mathcal{Q}$
13:   **if** $C_{sample}[t_s] \geq C_{train}[t_s]$ **then**
14:    **for** $j = 1$ to $t_s$ **do**
15:     $C_{train}[j] \leftarrow C_{train}[j] + 1$    ▷ Update train counts for all previous steps
16:    **end for**
17:    **yield** $t_s + 1$          ▷ Yield the steps to be trained
18:   **end if**
19:   $C_{sample}[t_s] \leftarrow C_{sample}[t_s] + 1$    ▷ Update sample count for the drawn step
20:  **end for**
21: **end while**

---

## A.5 Training Details of Online Replay Buffer

As in Algorithm 3, COAF employs an online replay buffer to implement multi-step training. When processing data at time step $t$, including atmospheric data at times $t$ and $t - \tau$, as well as oceanic data at time $t$, we take the atmospheric data at $t$ as April 5, 2025 (00:00 UTC) as an example.

In this case, the input atmospheric data includes April 5, 2025 (00:00 UTC) and April 4, 2025 (18:00 UTC), while the input oceanic data corresponds to April 4, 2025. The time embedding for ocean prediction is set to 0.25, indicating that we aim to generate oceanic data for six hours later.

During the first iteration of training, the model outputs atmospheric data for April 5, 2025 (06:00 UTC) and the predicted oceanic data six hours later. Since oceanic data is processed daily, only the atmospheric prediction error is calculated, and backpropagation is used to train the model. Subsequently, the output atmospheric data for April 5, 2025 (06:00 UTC) and the previously input atmospheric data for April 5, 2025 (00:00 UTC) are re-entered into the model, keeping the oceanic input unchanged. The time embedding for ocean prediction is set to 0.5, meaning that the model outputs atmospheric data for April 5, 2025 (12:00 UTC) and oceanic data for 12 hours later. At this point, only the atmospheric prediction error is calculated, and backpropagation is performed to train the model.

This process is repeated until the inputs are the atmospheric data for April 5, 2025, along with the original oceanic data (April 4, 2025). The time embedding for ocean prediction is set to 1, indicating that the model predicts oceanic data for one day ahead. The model then outputs atmospheric data for April 6, 2025 (00:00 UTC) and the oceanic data for April 6, 2025. At this point, both the atmospheric and oceanic losses are calculated and used for backpropagation to train the model.

This process continues, with oceanic losses used for training only when the ocean prediction time embedding is an integer. Otherwise, only atmospheric losses are used for training. The iteration count for this process is determined by sampling through the online replay buffer.

Additionally, if the initial data sampled includes atmospheric data for April 4, 2025 (18:00 and 12:00 UTC), as well as oceanic data for April 3, 2025, the next atmospheric prediction is for April 5, 2025 (00:00 UTC). At this point, the ocean prediction time embedding is set to 1, and the model outputs atmospheric data for April 5, 2025 (00:00 UTC) and oceanic data for April 4, 2025. Both oceanic and atmospheric losses are calculated and used to train the model via backpropagation.

In the next iteration of training, the predicted atmospheric data for April 5, 2025 (00:00 UTC), along with the atmospheric data for April 4, 2025 (18:00 UTC), and the initial oceanic data for April 3, 2025, are fed into the model. The ocean prediction time embedding is set to 1.25. At this point, only the atmospheric data is used for training, and the process continues iteratively.

# B Experiment

## B.1 Software and Hardware

The model is implemented in PyTorch. The training and inference are conducted on eight 80GB NVIDIA A100 devices. For Stage One, the training process takes 2.5 days using 8 NVIDIA A100 GPUs. Stage Two requires 1 day, while Stage Three takes 20 hours on 8 NVIDIA A100 GPUs.

## B.2 Metrics

We evaluated forecast performance using the latitude-weighted Root-Mean-Square Error (WRMSE) and the latitude-weighted Anomaly Correlation Coefficient (WACC). WRMSE is defined as follows:

$$\text{RMSE}(c, \tau) = \frac{1}{T} \sum_{i=1}^{T} \sqrt{\frac{1}{WH} \sum_{w=1}^{W} \sum_{h=1}^{H} \alpha(h)(x_{c,w,h}^{i+\tau} - \hat{x}_{c,w,h}^{i+\tau})^2}, \tag{16}$$

where $\alpha(h)$ represents the latitude weight defined as $W \cdot \frac{\cos(\alpha_{w,h})}{\sum_{w'=1}^{W} \cos(\alpha_{w',h})}$. Here, $c, w, h$ denote the indices for channel, latitude, and longitude, respectively.

---

**Algorithm 3** COAF Training with Online Replay Buffer

---

**Require:** Dataset $\mathcal{D}$, COAF model, Online Buffer Sampler $\mathcal{S}$
1: Initialize Optimizer
2: **for** each initial state $(A_{t-1}, A_t, O_t)$ from $\mathcal{D}$ **do**
3: $\quad$ $n \leftarrow \mathcal{S}$.next() $\qquad\qquad\qquad\qquad\qquad$ ▷ Get number of rollout steps from Online Buffer
4: $\quad$ Initialize states: $A_{curr} \leftarrow A_t, A_{prev} \leftarrow A_{t-1}, O_{input} \leftarrow O_t$

5: $\quad$ **for** $i = 1$ to $n$ **do** $\qquad\qquad\qquad\qquad$ ▷ Perform autoregressive rollout training for n steps
6: $\qquad\qquad\qquad\qquad\qquad\qquad\qquad$ ▷ — Ocean Prediction Step —
7: $\qquad$ $C_a \leftarrow \text{AtmosFluxEncoder}(A_{curr}, A_{prev})$
8: $\qquad$ $t'_{ocean} \leftarrow i \times \Delta t$ $\qquad\qquad\qquad\qquad$ ▷ Calculate continuous time step for ocean
9: $\qquad$ $O_{pred} \leftarrow \text{OceanModel}(O_{input}, t'_{ocean}, C_a)$
$\qquad\qquad\qquad\qquad\qquad\qquad$ ▷ — Atmosphere Prediction Step —
10: $\qquad$ $C_o \leftarrow \text{OceanFluxEncoder}(O_{pred})$
11: $\qquad$ $A_{next} \leftarrow \text{AtmosModel}(A_{curr}, A_{prev}, C_o)$
$\qquad\qquad\qquad\qquad\qquad\qquad\qquad\qquad$ ▷ — Loss Calculation —
12: $\qquad$ $\mathcal{L}_{air} \leftarrow \text{Loss}(A_{next}, A_{target,t+i})$
13: $\qquad$ $\mathcal{L}_{sea} \leftarrow 0$
14: $\qquad$ **if** $t'_{ocean}$ is a daily interval (e.g., $i \pmod 4 == 0$) **then**
15: $\qquad\qquad$ $\mathcal{L}_{sea} \leftarrow \text{Loss}(O_{pred}, O_{target,t+i})$
16: $\qquad$ **end if**
17: $\qquad$ $\mathcal{L}_{total} \leftarrow \mathcal{L}_{air} + \mathcal{L}_{sea}$
$\qquad\qquad\qquad\qquad\qquad\qquad$ ▷ — Update states for next iteration —
$\qquad\qquad\qquad\qquad\qquad\qquad$ ▷ Update model parameters after the rollout
18: $\qquad$ Update COAF parameters using $\nabla \mathcal{L}_{total}$
19: $\qquad$ $A_{prev} \leftarrow A_{curr}.detach()$
20: $\qquad$ $A_{curr} \leftarrow A_{next}.detach()$
21: $\quad$ **end for**
22: **end for**

---

WACC is defined as follows:

$$\text{ACC}(c, \tau) = \frac{1}{T} \sum_{i=1}^{T} \frac{\sum_{w,h} \alpha(h) x'^{i+\tau}_{c,w,h} \hat{x}'^{i+\tau}_{c,w,h}}{\sqrt{\sum_{w,h} \alpha(h)(x'^{i+\tau}_{c,w,h})^2 \sum_{w,h} \alpha(h)(\hat{x}'^{i+\tau}_{c,w,h})^2}}, \qquad (17)$$

where

$$x'^{i+\tau}_{c,w,h} = x^{i+\tau}_{c,w,h} - C^{i+\tau}_{c,w,h}, \qquad (18)$$

and

$$\hat{x}'^{i+\tau}_{c,w,h} = \hat{x}^{i+\tau}_{c,w,h} - C^{i+\tau}_{c,w,h}. \qquad (19)$$

Here, $C^{i+\tau}_{c,w,h}$ is the climatological mean over the day-of-year containing the validity time $i + \tau$ for a given weather variable $c$ at longitude $w$ and latitude $h$. It is averaged from the years 1993 to 2016 with the ERA5 data on a daily basis.

### B.3 COMPARISON WITH CONCAT TRAINING

To validate the necessity and effectiveness of the multi-stage approach in COAF, we processed the atmospheric data into daily averages and combined it with ocean data to train the model from scratch. The comparison results are shown in Table 5. For a fair comparison, we transformed the COAF prediction results into daily averages and calculated the metrics. The results indicate that directly concatenating ocean and atmospheric data performs significantly worse than COAF. This discrepancy is likely attributed to the limited amount of training data. Since the ocean dataset only contains daily data from 1993 to 2021, the concatenation of ocean and atmospheric data results in insufficient training samples, leading to poorer performance. In contrast, COAF separates the two datasets, allowing it to leverage the pre-trained atmospheric model and remain unaffected by the limited availability of ocean data.

Table 5: Comparison of COAF and Training of concat daily RMSE at lead times of 3, 5, 10 days in 2022.

| Method | Z500 | T850 | T2M |
|---|---|---|---|
| Training of Concat | 164/346/803 | 1.03/1.75/3.46 | 0.9/1.4/2.5 |
| COAF | **92/219/583** | **0.73/1.21/2.55** | **0.73/1.06/1.93** |

## B.4 THE VERSATILITY VERIFICATION OF COAF

To validate the versatility of COAF, we integrated the well-known Pangu-Weather model as the atmospheric model for training and testing. Noted that since Pangu-Weather is not fully open-sourced, we are only able to utilize the NVIDIA-reproduced version (`https://github.com/NVIDIA/physicsnemo/blob/main/physicsnemo/models/pangu/pangu.py`) for training and testing, which might lead to a performance gap between this version and their original paper. From Table 6, it can be seen that for other transformer-based models, COAF can still effectively achieve air-sea coupling and improve performance. Nevertheless, we are still able to verify the versatility of our COAF framework, because Pangu-Weather can be easily incorporated. We believed that once the official code of Pangu-Weather is integrated, its performance could also undergo an improvement.

Table 6: Comparison of COAF(Pangu-Weather) and Pangu-Weather RMSE at lead times of 3, 5, 10, 15 days in 2022.

| Method | Z500 | T850 | T2M |
|---|---|---|---|
| Pangu-Weather | 235/439/851/1052 | 1.79/2.67/4.37/5.3 | 1.73/2.17/3.15/3.71 |
| COAF(Pangu-Weather) | **195/370/698/813** | **1.4/2.0/3.2/3.7** | **1.51/1.84/2.60/2.95** |

## B.5 ABLATION STUDY ON UTILIZING CROSS-ATTENTION FOR COUPLING INTERACTION

In addition, we also attempted to use the cross-attention mechanism for air-sea coupling interactions. As shown in the Table 7, although the cross-attention method achieves some improvement compared to using only the atmospheric model, it still falls short compared to directly adopting the time-control method. This may be ascribed to that the cross-attention mechanism focuses more on high-resolution detail information, which is prone to larger errors during the iterative process. Furthermore, long-term air-sea interactions are more about macroscopic energy exchanges, which may explain why the performance of the cross-attention method is relatively worse.

Table 7: Comparison of Cross-Attn and Time-Control at lead times of 3, 5, 10, 15 days in 2022.

| Method | Z500 | T850 | T2M |
|---|---|---|---|
| Atmospheric-only model | 133/287/703/884 | 1.09/1.66/3.12/3.73 | 1.01/1.41/2.46/2.99 |
| Cross-Attn | 133/287/668/804 | 1.07/1.63/2.96/3.43 | 0.99/1.35/2.24/2.60 |
| Time-Control | **131/277/634/769** | **1.07/1.61/2.87/3.32** | **0.98/1.34/2.19/2.54** |

## B.6 ATOMOSPHERIC RESULTS

The latitude-weighted Anomaly Correlation Coefficient (WACC) measures a model's ability to predict deviations from normal conditions, with higher values indicating greater prediction accuracy. We compared the WACC of COAF with various models. Table 8 shows that COAF also demonstrates better performance in WACC for nearly all lead times and most variables. Additionally, we visualized the Z500 prediction results for 3-day, 5-day, and 10-day lead times, as shown in Figure 6. As the forecast step increases, the absolute error also increases and diffuses to adjacent areas.

In addition, we also compared COAF with Stormer, Climax, and WeatherGFT. The results (Figure 7) show that COAF performs comparably to Stormer in the short term and gains an advantage as the forecast period extends. This is because Stormer uses a multi-member ensemble forecast,

Table 8: Comparison of COAF, IFS and different AI models on WACC (higher is better) at lead times of 3, 5, 10, and 15 days.

| Method | Z500 | T850 | T2M | U10 | V10 |
|---|---|---|---|---|---|
| IFS-HRES | 0.99/0.93/0.51/* | 0.93/0.84/0.45/* | 0.93/0.87/0.62/* | 0.89/0.76/0.33/* | 0.89/0.75/0.29/* |
| Pangu-Weather | 0.99/0.93/0.53/0.21 | 0.94/0.87/0.46/0.19 | 0.94/0.87/0.60/0.40 | 0.91/0.78/0.34/0.14 | 0.90/0.78/0.30/0.10 |
| GraphCast_operational | 0.99/0.94/0.55/0.24 | 0.95/0.88/0.49/0.21 | 0.93/0.88/0.63/0.45 | 0.92/0.80/0.36/0.16 | 0.92/0.80/0.33/0.12 |
| FuXi | 0.99/0.94/0.63/0.39 | 0.95/0.88/0.57/0.34 | 0.95/0.89/0.71/0.60 | 0.92/0.81/0.48/**0.31** | 0.92/0.81/0.44/0.26 |
| COAF. | **0.99/0.94/0.63/0.39** | **0.95/0.89/0.58/0.35** | **0.95/0.90/0.72/0.60** | **0.93/0.83/0.49/0**.30 | **0.93/0.82/0.45/0.26** |

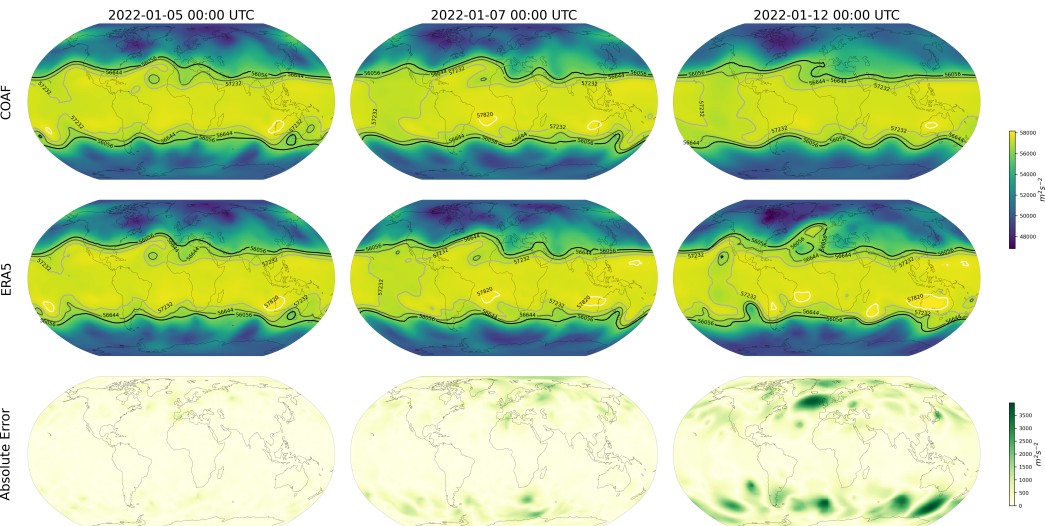

Figure 6: Visualization comparison of Z500 prediction results for 3-Day, 5-Day, and 10-Day forecasts at the initialization time of January 2, 2022, 00:00 UTC. The first row displays the model outputs, the second row shows the corresponding ERA5 states, and the third row illustrates the Absolute Error between the two.

offering better performance than single-member models. Compared to Climax and WeatherGFT, COAF maintains an overall lead. Climax, with its time-based direct prediction framework, faces higher training difficulty, while WeatherGFT focuses on short intervals, resulting in lower overall performance.

### B.7 OCEANIC RESULTS

For continuous-time ocean prediction models, since the original ocean data are daily averages, we are unable to quantify the accuracy of 6-hour interval predictions. Therefore, we visualized the predicted and actual values of the lateral flow speed at a specific location near the equator in the ocean, as in Figure 8 which shows gradual changes with prediction time, indicating COAF will not introduce large errors for intervals smaller than 24 hours, thus leading to faithful predictions in 24 hours.

In addition to the simple comparison presented in Table 2 of main text, we also visualize various variables. The experimental results highlight the forecasting capabilities of the ocean prediction module of COAF across multiple variables and spatiotemporal scales. Figure 9 illustrates a comparative analysis of sea surface height (zos) visualizations for 3-day, 5-day, and 10-day forecasts initialized at 00:00 UTC on July 1, 2022. The sea surface height forecasts exhibit coherent patterns of large-scale features through the 10-day predictions, although there is a noticeable error growth, especially in the regions of western boundary currents.

Figure 10 focuses on temperature at a depth of 0.494 m, also initialized on July 1, 2022. Similar trends are observed in the model outputs for the 3-, 5-, and 10-day forecasts. The 10-day forecasts exhibit strong alignment with GLORYS12V1 states, particularly in the equatorial regions and warming pool areas. Additionally, COAF provides relatively accurate 10-day forecasts even amidst

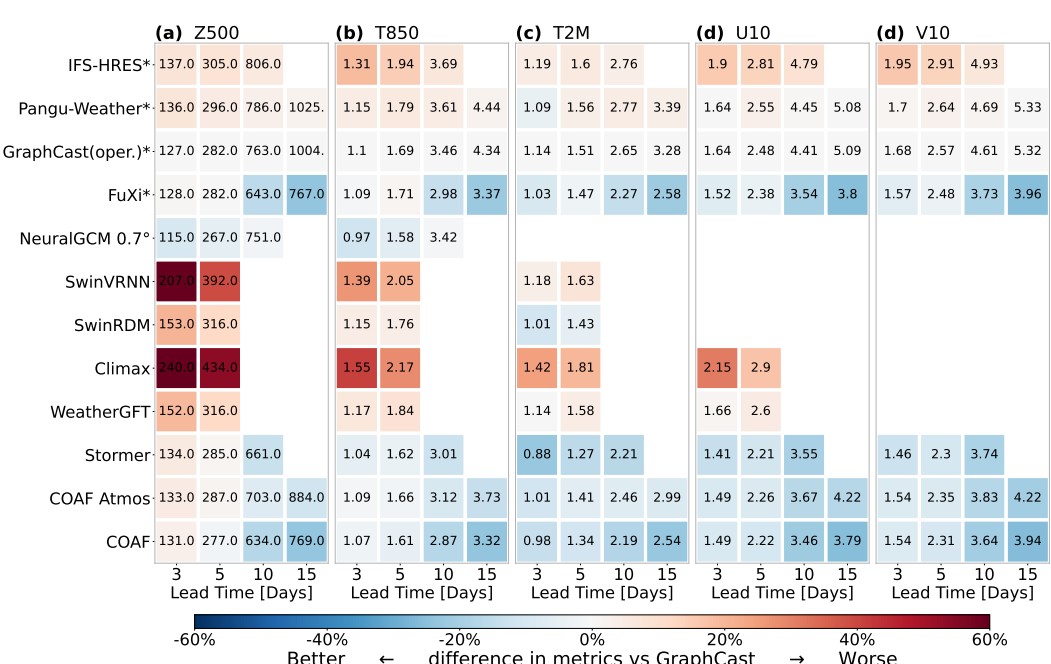

Figure 7: Comparison of more models.

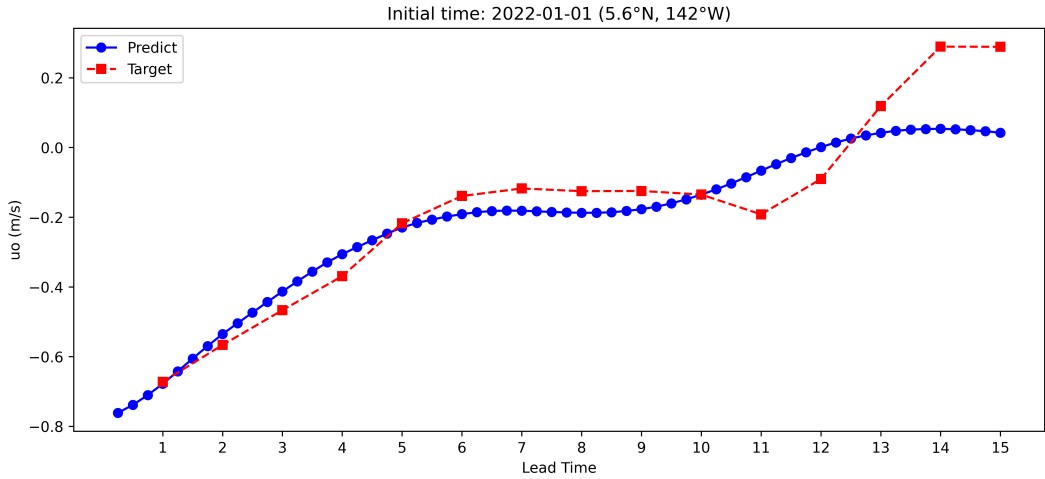

Figure 8: Visualization of the 6-hourly forecasts and actual values of uo within 15 days starting from January 1, 2022, at 5.6°N, 142°W.

high-frequency variations in current speeds and salinity, effectively revealing nutrient transport and ecological dynamics (see Figure 11).

Finally, to validate the key three-dimensional vertical structure of the ocean, we visualized the vertical meridional profile of upper ocean temperature at 180°E on July 11, 2022 (Figure 12). The results indicate that even on the tenth day, the model accurately predicts the ocean's three-dimensional thermal stratification. Furthermore, the prediction of the thermocline remains stable, accurately forecasting the position of the 20°C isotherm. This highlights the physically consistent predictive capability of the ocean model.

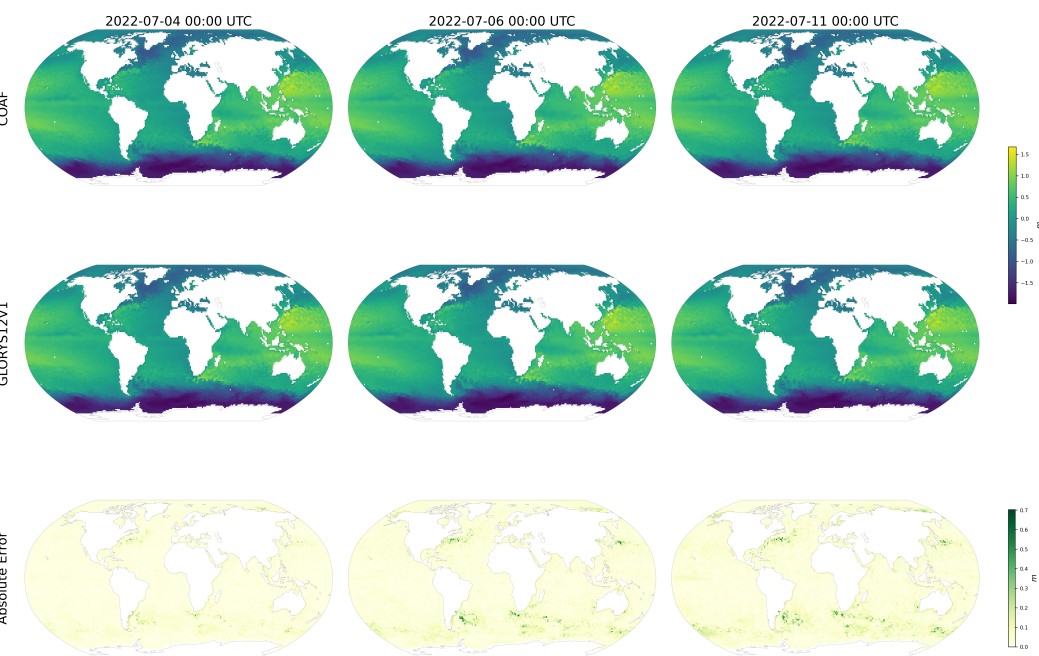

Figure 9: Comparison of Sea surface height (zos) visualizations from 3-, 5-, and 10-day forecasts initialized on 1 July 2022 00:00 UTC. The first row displays the model outputs, the second row shows the corresponding GLORYS12V1 states, and the third row illustrates the MAE between the two.

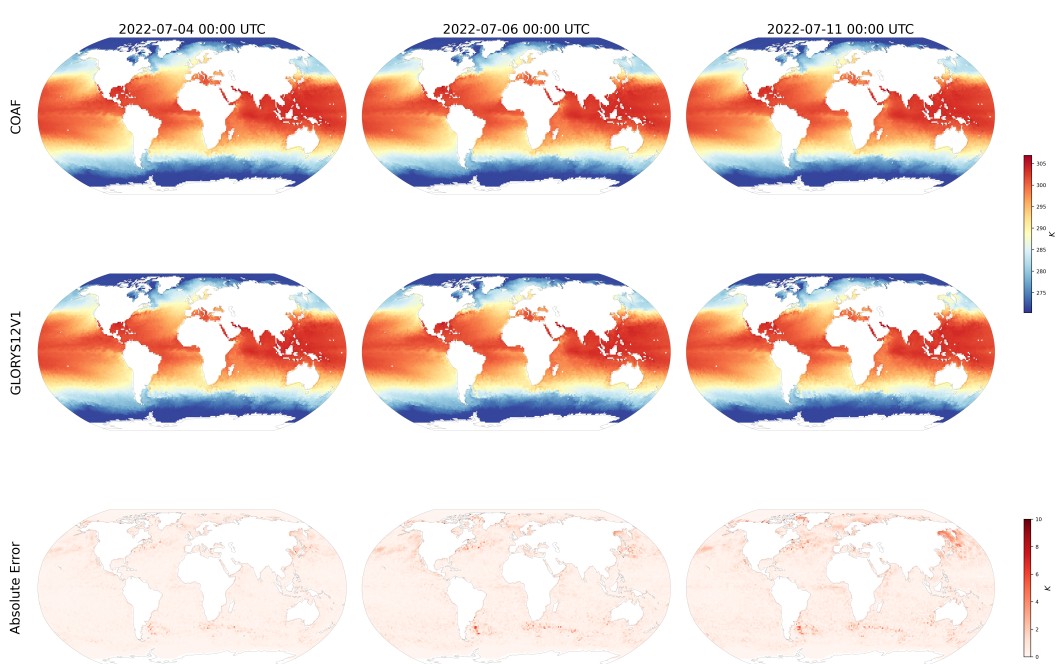

Figure 10: Comparison of thetao visualizations at a depth of 0.494 m from 3-, 5-, and 10-day forecasts initialized on 1 July 2022 00:00 UTC. The first row displays the model outputs, the second row shows the corresponding GLORYS12V1 states, and the third row illustrates the absolute MAE difference between the first two rows.

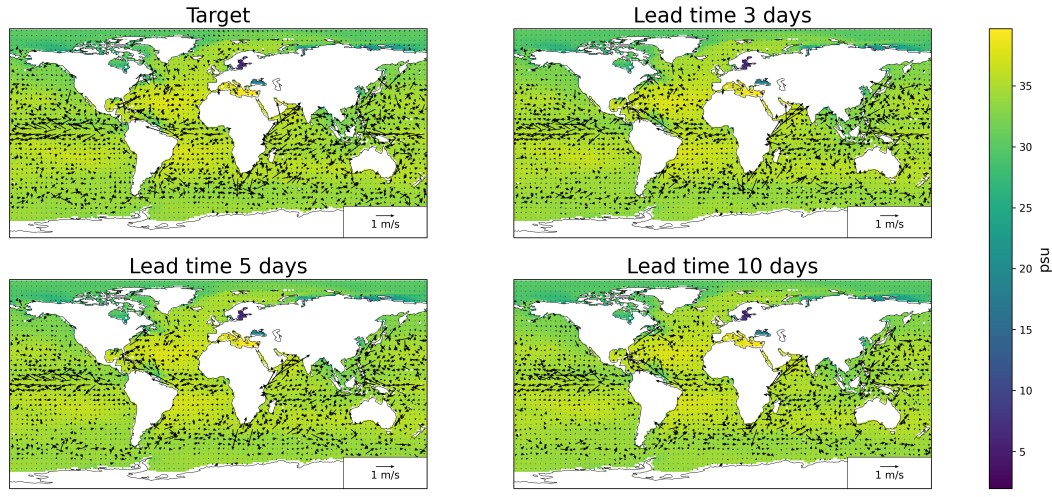

Figure 11: Visualisations of salinity and ocean current velocity forecasts valid for 11 July 2022 00:00 UTC with 3-day, 5-day, and 10-day lead times. The vectors represent the current velocity at the surface, while the shading shows salinity.

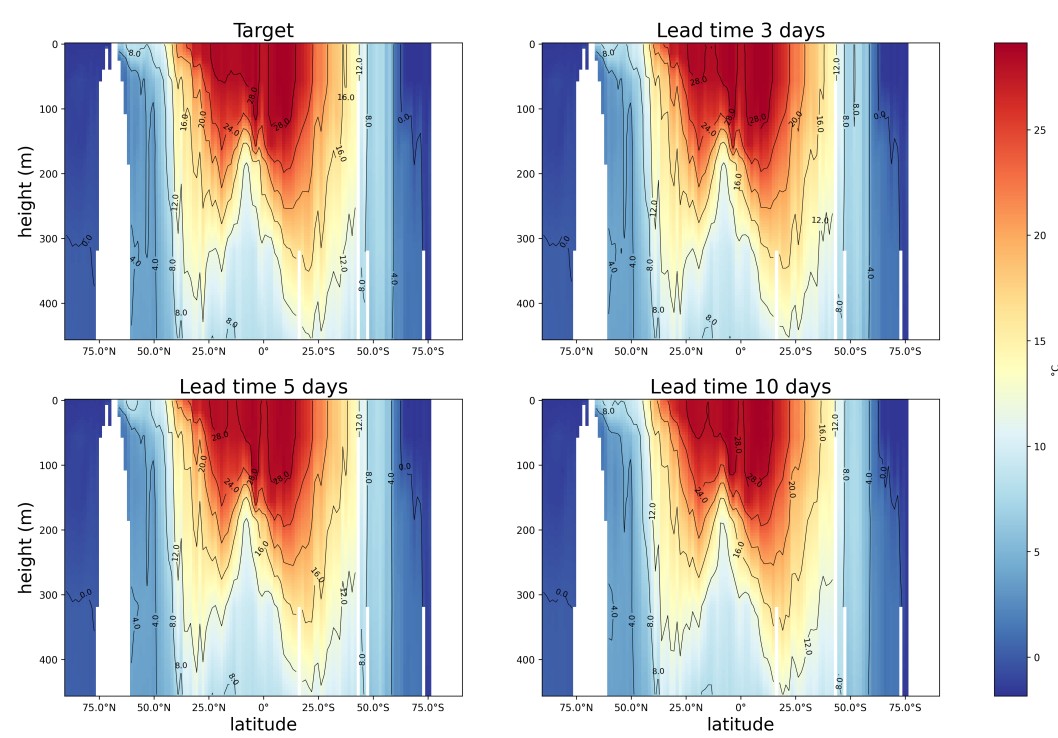

Figure 12: Vertical-meridional profile of upper ocean temperature forecast at 180°E for 11 July 2022 at 00:00 UTC across 3, 5, and 10-Day lead time.

