# OpenReview forum: "Bridging Oceans and Atmosphere: A More Comprehensive Weather Model"
_ICLR.cc/2026/Conference — ICLR 2026 Conference Withdrawn Submission_

### Official Review · Reviewer_kqfH · 2025-10-20

**Soundness:** 2
**Presentation:** 2
**Contribution:** 2
**Rating:** 2
**Confidence:** 3

**Summary:**

This paper proposes COAF (Coupled Ocean-Atmosphere Framework), a plug-and-play deep learning architecture designed to bridge oceanic and atmospheric forecasting models through learned flux exchange and temporal alignment mechanisms. COAF couples pre-trained atmospheric models (autoregressive) with continuous-time oceanic models using a Flux Generation Module that encodes each system’s state into a 1D “flux vector,” serving as a learned analog of physical energy–momentum exchange. A Conditional Transformer Coupler modulates these fluxes across domains, while an Online Replay Buffer dramatically reduces memory usage during long-term training.

**Strengths:**

- The framework addresses a scientific gap: integrating ocean and atmosphere representations, via a learned “flux” channel that emulates physical coupling. This approach is modular and compatible with existing pretrained models.

- The paper demonstrates consistent RMSE/WACC improvements across key variables and lead times, supplemented by clear ablations (no-ocean variant, flux module off, etc.). The results, especially at >10-day leads, are meaningful for operational forecasting.

**Weaknesses:**

- COAF mostly integrates known techniques, conditional modulation, replay buffers, and Swin-based encoders, into a multi-model coupling setup. The “flux generation” mechanism, while framed as novel, is conceptually similar to cross-attention or feature fusion methods widely used in multimodal learning.

- Ultimately, the lead time is too small for a coupled model. It should go beyond weather-scale, and into the subseasonal, seasonal, and climate scale. Otherwise, existing AINWP are well-suited for the task. The purpose of coupling should be to extend the timescale of forecasts.

- Phrases like “establishes a new paradigm for Earth system modeling” overreach the evidence presented. The method has yet to demonstrate meaningful coupling feedbacks, scalability, or robustness across seasons and regions. Existing coupled ESM ML emulators already exists and should be at least benchmarked.

- While the paper is thorough, it is overly dense with training algorithms (multi-stage coupler, online replay buffer pseudocode) that obscure intuition. The main innovation, how flux vectors actually mediate temporal mismatch, is buried under implementation details. I suggest the authors to clarify this part and frame it as the main contribution even more.

**Questions:**

See my weakness section above.

---

### Official Review · Reviewer_rqBn · 2025-10-27

**Soundness:** 1
**Presentation:** 2
**Contribution:** 2
**Rating:** 2
**Confidence:** 2

**Summary:**

The submission presents a novel, flexible, memory efficient framework (COAF) to couple the atmosphere and ocean component of the earth system in order to improve medium-range weather prediction. The atmospheric component uses an autoregressive backbone at a 6-hourly timestep while the oceanic component uses a neural model to couple to the atmospheric component every four timesteps.

The coupling is accomplished using a conditional transformer parameterized version of the atmosphere-ocean coupler of Ly (1995) where the atmosphere-to-ocean and ocean-to-atmospheric fluxes are used to update the atmospheric and oceanic state.

The main contributions of the work lie in the novelty of the effort to couple the two systems, as most state-of-the-art prediction models focus on atmospheric evolution only, neglecting the importance of oceanic evolution on medium-range weather forecasting.

**Strengths:**

The central idea that the paper addresses, i.e., atmosphere-ocean coupling through novel deep learning architectures, is an important one. I appreciate the authors' efforts in this direction. The approach is inspired from the traditional approach to couple the two systems and the use of Online Replay Buffer and three-staged training procedure could be of use to future studies in developing memory efficient forecasting systems.

**Weaknesses:**

1) I find the comparison presented in the study very inconsistent. In particular:
(a) The introduction is not clearly written, as it seems to make misleading statements in multiple places (see detailed point below)
(b) The comparison is a bit unorganized (see comments below and the Questions section)
(c) There are logical inconsistencies in multiple numbers provided

2) The Introduction seems to glorify/overhype ML-based weather forecasting while at the same time undermining traditional weather forecasting (as has seemingly become a trend in AI-weather forecasting cliques). The authors seem to forget that the current ML models are trained on global data generated by traditional models, and these models are pretty robust and based on solid mathematics and numerical computing frameworks. For instance, Lines 51 to 55, the authors seem to point out the problems with traditional forecasting. Moreover, the text goes on to blindly (over)-cite all the AI-weather forecasting models out there to make a minor point. That is bad writing. This is worsened by a heavy lack of citations of traditional work (related to the HRES model on Line 103.) Even Line 49 could use more references, including Stockdale et al. (1997), Xue et al. (2020), and Zhou and Harris (2022). Similarly, on L135, the Srinivas et al. and Rainaud et al. references are very vague ones. I would recommend that the authors add more appropriate citations that discuss how the atmos-ocean coupling is accomplished in ECMWF or NCAR's models.


References:

Zhou, L., & Harris, L. (2022). Integrated dynamics-physics coupling for weather to climate models: GFDL SHiELD with in-line microphysics. Geophysical Research Letters, 49, e2022GL100519. https://doi.org/10.1029/2022GL100519

Xue, P., Malanotte-Rizzoli, P., Wei, J., & Eltahir, E. A. B. (2020). Coupled ocean-atmosphere modeling over the Maritime Continent: A review. Journal of Geophysical Research: Oceans, 125, e2019JC014978. https://doi.org/10.1029/2019JC014978

Stockdale, T. N., 1997: Coupled Ocean–Atmosphere Forecasts in the Presence of Climate Drift. Mon. Wea. Rev., 125, 809–818, https://doi.org/10.1175/1520-0493(1997)125<0809:COAFIT>2.0.CO;2.

**Questions:**

Major:
1) The ocean model and variables are not described in sufficient detail.
2)  It appears that the FuXi model provides a better forecast skill than COAF Atmos and very similar performance to COAF. This makes me question the working hypothesis that coupling atmosphere and ocean, as presented in this study, leads to improved medium-range prediction. Can the authors clarify why this is the case?
3) Can you explain the rationale behind using GraphCast as a reference in Figure 3, but using FourCastNet in Table 2? To my understanding, ECMWF's AIFS (Lang et al. 2024) is the current state of the art, as has been discussed in multiple studies (Lang et al. (2024), Gupta et al. (2025)). I believe these (and other studies) should be references to make a clear point what justifies using GraphCast as the reference here.
4) Is there any special provision to distinguish between the ocean and sea-ice component of the Earth system? There does not seem to be any such provision, which can lead to inaccuracies in A2O and O2A fluxes.
5) Table 1: Given that FuXi and COAF offer similar performance, I would recommend adding FuXi to Table 1 as well.
6) Also, I think Table 1 is misleading, as many models mentioned in the table were trained at a 0.25-degree resolution. Which makes them 5-6 times finer. This resolution corresponds to a 25-36 times increase in model size and thus training times. Moreover, these models also considered a much broader range of input variables and thus have more channels. Thus, accounting for this, I don't think the COAF Stage 3 training time itself offers any significant improvement in training times.
7) I do not know what to make of Figure 4. It does not inform me of anything, I am not sure why all of a sudden the authors compare their prediction to ERA5 (and not HRES), and why no other model has been shown in the figure (GraphCast for instance). In its current state, I would rather just remove this figure and it does not provde any useful information whatsoever.
8) Fig 3 and Fig 4 suggest to me that even including ocean-atmos coupling does not provide any notable improvements over medium-range timescales - making me question the central hypothesis behind the analysis. Can the authors comment on why that is?
9) Fig 6: The large-scale features in geopotential height are arguably much easier to predict. The figure clearly shows that the COAF models fails to capture the small-scale features both at 5 day and 10 day lead times. This further reduces my confidence in the study and the "SOTA" forecasting skill reported in this study.


Minor:
1) L149: I don't think such explorations remain unexplored. There have been ample 'traditional' explorations.
2) L363: HRES is NOT a dynamical core but a proper operational forecast model built on top of the spectral dynamical core
3) Line 328-329: over 13 pressure levels, not along with.
4) In Fig 3, IFS-HRES should not have a * because the original resolution for HRES is 9 km.
5) Why are ocean variables not shown in the main text? Given that the central focus of the study is on atmos-ocean coupling?
6) L478-479: COAF's performance cannot possibly be claimed as SOTA when it is not even compared against the actual SOTA (which is AIFS). Why not compare the prediction skill against ECMWF's AIFS instead, which is the actual state of the art in ML weather prediction right now?

References:
1) Lang, Simon, et al. "AIFS--ECMWF's data-driven forecasting system." arXiv preprint arXiv:2406.01465 (2024).

2) Gupta, Aman, Aditi Sheshadri, and Dhruv Suri. "MAUSAM: An Observations-focused assessment of Global AI Weather Prediction Models During the South Asian Monsoon." arXiv preprint arXiv:2509.01879 (2025).

3) Noah D. Brenowitz, Yair Cohen, Jaideep Pathak, Ankur Mahesh, Boris Bonev, Thorsten Kurth, Dale R.
Durran, Peter Harrington, and Michael S. Pritchard. A Practical Probabilistic Benchmark for AI Weather
Models. Geophysical Research Letters, 52(7):e2024GL113656, 202

4) Rasp, Stephan, et al. "WeatherBench 2: A benchmark for the next generation of data‐driven global weather models." Journal of Advances in Modeling Earth Systems 16.6 (2024): e2023MS004019.

---

### Official Review · Reviewer_GFjC · 2025-10-29

**Soundness:** 3
**Presentation:** 3
**Contribution:** 2
**Rating:** 2
**Confidence:** 3

**Summary:**

This paper introduces a deep learning architecture designed for medium-range global weather forecasting by explicitly coupling atmospheric and oceanic dynamics. The framework leverages pre-trained atmospheric and oceanic models and connects them using a flux-generation module that handles the temporal and spatial mismatches between datasets. COAF implements an online replay buffer for memory-efficient long-term training and introduces a conditional interaction mechanism for effective cross-domain information exchange, minimizing changes to existing model architectures.

While the paper is overall sound, its novelty is mainly to domain-specific methodologies relevant to Earth-system modeling. I am therefore doubting whether ICLR is the right venue to present these results, which explains my grade.

**Strengths:**

- The paper addresses the temporal scale mismatch between atmospheric and oceanic data when building coupled Earth system models. The three-stage training approach allows leveraging pre-trained atmospheric models without retraining from scratch and reduces computational costs compared to competitors like Pangu-Weather or GraphCast.
- The online replay buffer reduces memory consumption by over four orders of magnitude (from 4.6 GB to 0.08 MB) while maintaining comparable performance to standard replay buffers, making the approach more scalable to higher resolutions. Performance improvements are consistent: approximately 10% reduction in RMSE for key variables (Z500, T2m) beyond 10-day lead times compared to uncoupled baselines.
- A novel flux generation module that compresses spatial features to couple  the 2 components.

**Weaknesses:**

- None of the aforementioned methods is particularly novel on its own. Given that ICLR is primarily a ML conference I wonder, whether these results should be presented here.

- The experiments are conducted at 1.4° resolution while state-of-the-art models operate at 0.25°. The authors justify this as isolating the coupling effect, but it significantly limits practical relevance and direct comparison with leading methods.

- The ocean component uses 5 depth-based expert models covering 13 vertical levels. This partitioning seems arbitrary and lacks ablation studies showing why 5 experts is optimal. The continuous-time ocean modeling, while addressing the temporal mismatch, introduces potential interpolation errors between daily observations that aren't quantified.​

- The flux generation module compresses spatial features to a 1D vector through attention pooling. This aggressive dimensionality reduction may discard localized coupling information that matters for regional phenomena. No analysis demonstrates what information is preserved or lost in this bottleneck.​

- Evaluation focuses primarily on global metrics (WRMSE, WACC). Regional performance, particularly in key coupling regions like tropical oceans or boundary currents, receives minimal analysis. The visualization of ocean currents shows accuracy degradation in western boundary current regions, but this isn't explored systematically.​

- The comparison baseline "COAF_Atmos" is just their own atmospheric model without ocean coupling. Comparisons with other coupled models is limited, leaving unclear how COAF performs against actual coupled competitors at comparable timescales. OLA and Dlesym are cited but I cannot find any comparison with these models.

**Questions:**

- Why isnt there a comparison to other coupled models such as OLA or Dlesym?

---

### Note · Authors · 2026-01-06

I have read and agree with the venue's withdrawal policy on behalf of myself and my co-authors.